

SciPost Phys. Lect. Notes 9 (2019)

# On model selection in cosmology

**Martin Kerscher[1⋆] and Johen Weller[1,2,3]**

**1** Universitäts-Sternwarte Ludwig-Maximilians-Universität München,
Scheinerstr. 1, 81679 München, Germany
**2** Max Planck Institute for Extraterrestrial Physics,
Giessenbachstrasse, 85748 Garching, Germany
**3** Excellence Cluster Origins,
Boltzmannstr. 2, 85748 Garching, Germany

⋆ martin.kerscher@lmu.de

## Abstract

We review some of the common methods for model selection[1]: the goodness of fit, the likelihood ratio test, Bayesian model selection using Bayes factors, and the classical as well as the Bayesian information theoretic approaches. We illustrate these different approaches by comparing models for the expansion history of the Universe. In the discussion we highlight the premises and objectives entering these different approaches to model selection and finally recommend the information theoretic approach.

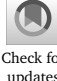
## Contents

---

[1]The expression "model selection" sometimes "model choice" is well established. In physics "model comparison" is presumably more appropriate, but we will stick with the de facto standard.

# 1   Introduction

In science we often have several competing theoretical models which try to explain the same natural phenomenon. Based on measured data we want to decide which model is the better one. As an example we consider two different cosmological models, a cold dark matter model (CDM) with a cosmological constant $\Lambda$ called $\Lambda$CDM, and a cold dark matter model with a constant equation of state $p = w\varrho$ for the dark energy component called $wCDM$. With both models we try to explain observations like the cosmic microwave background, galaxy cluster counts, supernovae distance measurements, to name only a few. Models with more parameters typically allow for a closer fit of the data, but are such models with more parameters indeed better (see Fig. 1)? In such a context one often refers to Ockham's razor that one should not introduce additional parameters if they are not needed[2]. One task of model selection is to make this statement quantitative.

To set a well defined stage, consider some measured data points $d_i = (x_i, y_i)$ with $i \in \{1, \ldots, N\}$. A model is providing a function $f(x, \boldsymbol{\theta})$ such that $f(x_i, \boldsymbol{\theta})$ is approximating $y_i$ for each $i$. The parameters $\boldsymbol{\theta} = (\theta_1, \ldots, \theta_K)$ are from $A \subset \mathbb{R}^K$. For simplicity we assume that $x, x_i, y_i \in \mathbb{R}$ and also $f(x, \boldsymbol{\theta})$ is real valued[3]. The best fitting parameters $\boldsymbol{\theta}^\star \in A \subset \mathbb{R}^K$ of the model are then determined from a Bayesian approach, a maximum likelihood procedure, or a simple least–square–fit. If one considers only a single model and has a good idea about the priors for the parameters, then many physicists would agree that a Bayesian parameter estimation procedure is the appropriate thing to do (see [2]).

The situation is more complicated if one considers at least one other model $g(x, \boldsymbol{\phi})$ with parameters $\boldsymbol{\phi} \in B \subset \mathbb{R}^L$. For simplicity we name the models after the functions $f$ and $g$. Typically the dimensions of the parameter spaces differ $L \neq K$ and also the parameter spaces may not overlap. We can determine the optimal parameters $\boldsymbol{\theta}^\star$ and $\boldsymbol{\phi}^\star$ for each of the models $f$ and $g$. But the question remains, which of the models is "better". This situation is illustrated in Fig. 1.

As a starting point we will briefly discuss some of the common methods used for parameter estimation. Then we will present methods used for the selection of models and also comment on the approximations and numerical methods used. In section 3 we use these methods to compare two models for the expansion history of the Universe. In the discussion we highlight the premises and objectives entering the different approaches and recommend the information theoretic procedure, preferably in its Bayesian flavour. In appendix A we summarise properties of statistical tests, the empirical distribution function, and the Kullback-Leibler divergence. In

---

[2]Numquam ponenda est pluralitas sine necessitate. Attributed to William of Ockham and sometimes earlier to Duns Scotus. See Thorburn [1] for an historical account.

[3]Choosing a real valued $f$ and real $x, x_i, y_i$ is done for notational simplicity. We could choose a more complex mapping without touching the following arguments.

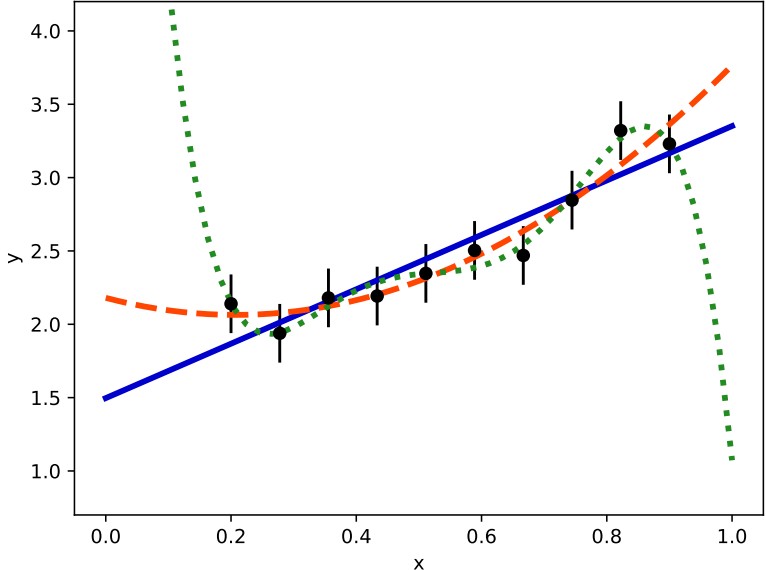

Figure 1: Three polynomials fitted to 10 data points: $f(x, \boldsymbol{\theta}) = \theta_0 + \theta_1 x$ (solid blue line), $g(x, \boldsymbol{\phi}) = \phi_0 + \phi_1 x + \phi_2 x^2$ (dashed red line) $h(x, \boldsymbol{\psi}) = \psi_0 + \psi_1 x + \psi_2 x^2 + \psi_3 x^3 + \psi_4 x^4 + \psi_5 x^5$ (dotted green line). Which of these polynomials is the "best" model (see also Munroe [3])?

appendix B we provide some details of the numerical implementation and in appendix C we discuss the application and especially the error budget in more detail. We assume some familiarity with statistical methods used in physics and cosmology (see for example [4, chap. 40] for a short review). A comprehensive introduction to statistics including model selection is Wassermann [5]. An introduction to model selection with a focus on the information theoretic approach is Burnham & Anderson [6]. Most of the material shown here is not new. Some of the results are scattered throughout the literature so we try to present them in a coherent fashion and give due reference.

## 1.1 Parameter estimation

In the following we will give a short review of different methods for parameter estimation. The starting point for a comparison of a model with the data is in most cases the likelihood. To specify the likelihood function $p_f(\boldsymbol{d} \,|\, \boldsymbol{\theta})$, the model itself $f(x, \boldsymbol{\theta})$ and an error model for the data is needed. The vector of measurements is $\boldsymbol{d} = (x_i, y_i)_{i=1}^N$ and $p_f(\boldsymbol{d} \,|\, \boldsymbol{\theta})$ is the probability of obtaining the measured data points $\boldsymbol{d}$, given the parameters $\boldsymbol{\theta}$ in the model $f$. Often one assumes Gaussian errors, then the likelihood reads

$$p_f(\boldsymbol{d} \,|\, \boldsymbol{\theta}) = \frac{1}{((2\pi)^N \det(\Sigma))^{\frac{1}{2}}} \, \exp\left[ -\tfrac{1}{2}(\boldsymbol{y} - \boldsymbol{f}(\boldsymbol{x}, \boldsymbol{\theta}))^T \Sigma^{-1} (\boldsymbol{y} - \boldsymbol{f}(\boldsymbol{x}, \boldsymbol{\theta})) \right], \tag{1}$$

with $\boldsymbol{y} = (y_1, \ldots, y_N)^T$, $\boldsymbol{x} = (x_1, \ldots, x_N)^T$, $\boldsymbol{f}(\boldsymbol{x}, \boldsymbol{\theta}) = (f(x_1, \boldsymbol{\theta}), \ldots, f(x_N, \boldsymbol{\theta}))^T$ and the covariance matrix $\Sigma$. With a maximum likelihood estimator we determine the parameters $\boldsymbol{\theta}^\star$ which are maximising $p_f(\boldsymbol{d} \,|\, \boldsymbol{\theta}^\star)$. Hence in choosing the parameter $\boldsymbol{\theta}^\star$, the data points $\boldsymbol{d}$ become the most probable data points given the model $f$.

The least square method is a simplified version of the maximum likelihood estimator [7]. The likelihood is assumed to be Gaussian with a diagonal $\Sigma$ and the $\sigma_i$'s on the diagonal.

Searching the maximum of $p_f(\boldsymbol{d}\,|\,\boldsymbol{\theta})$, or of

$$\log p_f(\boldsymbol{d}\,|\,\boldsymbol{\theta}) = -\tfrac{N}{2}\log(2\pi) - \tfrac{1}{2}\log(\det(\Sigma)) - \tfrac{1}{2}\sum_{i=1}^{N}\frac{(y_i - f(x_i,\boldsymbol{\theta}))^2}{\sigma_i^2} \tag{2}$$

gives the same result as searching for the minimum of

$$\chi_f^2 = \sum_{i=1}^{N}\frac{(y_i - f(x_i,\boldsymbol{\theta}))^2}{\sigma_i^2}. \tag{3}$$

This minimum determines the best parameters $\boldsymbol{\theta}^*$ of the least square fit.

In a Bayesian setting we also need the prior distribution $p_f(\boldsymbol{\theta})$ of the parameters for model $f$. Using Bayes theorem we can determine the posterior distribution

$$p_f(\boldsymbol{\theta}\,|\,\boldsymbol{d}) = \frac{p_f(\boldsymbol{d}\,|\,\boldsymbol{\theta})\,p_f(\boldsymbol{\theta})}{p_f(\boldsymbol{d})}, \tag{4}$$

the distribution of the parameters $\boldsymbol{\theta}$, given the data $\boldsymbol{d}$ and the model $f$. Contrary to the posterior distribution $p_f(\boldsymbol{\theta}\,|\,\boldsymbol{d})$, the likelihood $p_f(\boldsymbol{d}\,|\,\boldsymbol{\theta})$ is the distribution of the data $\boldsymbol{d}$ given the parameters $\boldsymbol{\theta}$ of the model $f$. The normalisation $p_f(\boldsymbol{d})$ is called the evidence or marginal likelihood. The evidence can be obtained by an integration in parameter space

$$p_f(\boldsymbol{d}) = \int p_f(\boldsymbol{d}\,|\,\boldsymbol{\theta})\,p_f(\boldsymbol{\theta})\mathrm{d}\boldsymbol{\theta}. \tag{5}$$

Given the data and the model, the evidence is a normalisation constant of the posterior distribution. Hence, we do not need to calculate the evidence if we want to determine the maximum $\boldsymbol{\theta}^{\times}$ or the mean of the posterior distribution $p_f(\boldsymbol{\theta}^{\times}\,|\,\boldsymbol{d})$ only. The maximum[4] $\boldsymbol{\theta}^{\times}$ is called the maximum posterior (MAP) estimate. Clearly, there is more to parameter estimation than we covered here. We did not discuss how to choose priors, how to deal with nuisance parameters, or how to determine confidence or credibility regions. Nevertheless we provided the necessary prerequisites to be able to discuss model selection.

## 2 Model selection

Quite a few methods for model selection have been developed. Cosmological and astrophysical oriented reviews and books are for example [8], [9] [10]. A more philosophically inclined introduction with basic examples can be found in Sober [11, chapter 1].

### 2.1 The goodness of fit

The so called "goodness of fit" may serve as a starting point for this discussion, since it is often the first method students of physics learn in their lab–courses. One calculates the so called reduced $\chi_{f,\mathrm{red}}^2$

$$\chi_{f,\mathrm{red}}^2 = \frac{\chi_f^2}{n_{\mathrm{df}}}, \tag{6}$$

where $\chi_f^2$ is calculated using eq. (3) with the best fit parameters $\boldsymbol{\theta}^*$ of the model. $n_{\mathrm{df}}$ is the number of degrees of freedom, typically $n_{\mathrm{df}} = N - K$, with $N$ the number of data points and

---

[4]The different "best" parameter estimates receive different stars as labels: $\boldsymbol{\theta}^*$ (least–square) $\boldsymbol{\theta}^{\star}$ (maximum–likelihood), $\boldsymbol{\theta}^{\times}$ (MAP).

$K$ the number of parameters in the model $f$. If $\chi^2_{f,\text{red}} \approx 1$, this is considered a good fit, if $\chi^2_{f,\text{red}} > 1$ a bad fit and if $\chi^2_{f,\text{red}} < 1$ an overfit. Remember, one starts with a maximum likelihood estimate and makes assumptions about the data- and error-model which are often stark oversimplifications. As a remedy the number of degrees of freedom is determined from the "effective" number of independent data points. The problems of this approach are summarised in [12].

Some of the motivation for using the $\chi^2_f$ derives from the theory of statistical tests (see [13] or [10]). The $p$–value used in these tests is calculated as

$$p = 1 - G_{n_{\text{df}}}(\chi^2_f), \tag{7}$$

with $G_{n_{\text{df}}}$ the cumulative probability distribution function of a $\chi^2$ distributed random variable with $n_{\text{df}}$ degrees of freedom (see appendix A). Clearly $p$ is one-to-one with $\chi^2_f$. The $p$–value indicates how incompatible the data are with our null hypothesis (our model including the error model, [14]). The smaller the $p$–value, the greater is the statistical incompatibility of the data with the null hypothesis. Hence, given the model (the null hypothesis), a small $p$–value allows us to reject the model using a statistical test. From the $p$–value however we do not learn anything about the false negative rate (see appendix A). The $p$–value is a statement about data in relation to a specified hypothetical explanation (our model), not about the data itself, and especially not about other models. See the recent statement of the American Statistical Association on the (restricted) applicability of $p$–values [14].

Whether an alternative hypothesis/model is needed, was one of the issues in the debate about hypothesis testing between Fisher on one side and Neyman and Pearson on the other side (see Lehmann [15] for a breakdown of the arguments). With respect to the importance of the false negative rate and the specification of an alternative hypothesis we side here with Neymann and Pearson (see the following section 2.2).

## 2.2 Likelihood ratio test

For the selection of models Neyman and Pearson [16] developed the likelihood ratio test. As usual we first consider so called nested models. The model $f$ with parameter space $A$ is a special case of the model $g$ with parameter space $B$. More formally $A \subsetneq B$ and $f|_A \equiv g|_A$ restricted to $A$. Now we determine the best fitting parameter $\boldsymbol{\theta}^\star \in A$, and $\boldsymbol{\phi}^\star \in B$ and calculate

$$L = \frac{p_f(\boldsymbol{d} \mid \boldsymbol{\theta}^\star)}{p_g(\boldsymbol{d} \mid \boldsymbol{\phi}^\star)}. \tag{8}$$

Our null hypothesis is "$f$ is the true model with $\boldsymbol{\theta}^\star \in A$". The alternative is "$g$ is the true model with $\boldsymbol{\phi}^\star \in B$ but $\boldsymbol{\phi}^\star \notin A$". With these maximum likelihood estimates $\boldsymbol{\theta}^\star$, and $\boldsymbol{\phi}^\star$ we calculate $L$. Fixing a significance level $0 < \alpha < 1$ one can proceed and specify the test. Often one relies on Wilk's theorem [17]: for nested models and for large sample sizes $N$ the $\lambda = -2\log L$ is approximately $\chi^2$–distributed with the number of degrees of freedom equal to $\nu = \dim(B) - \dim(A)$. The $p$–value is calculated as $p = 1 - G_\nu(\lambda)$. We reject our null hypothesis if $p < \alpha$ with a predefined significance level $\alpha$ (see the appendix A). Contrary to the situation discussed with the goodness of fit, the alternative hypothesis is fully specified in the likelihood ratio test. The false negative rate[5] is the probability that the true alternative hypothesis (our model g) gets rejected. The Neyman-Pearson–Lemma [16] tells us that a test based on the likelihood ratio is minimising the false negative rate. In this sense the likelihood ratio test is optimal.

---

[5]The false negative rate is also called the type II or $\beta$ error.

In the introduction we already considered a more general, non-nested setting. Vuong [18] discusses the likelihood ratio test for overlapping or non-nested models and he derives the relevant limiting distribution (not necessarily a $\chi^2$-distribution anymore). The application of the likelihood ratio test in this more general setting is reviewed by Lewis et al. [19].

### 2.3 Bayesian model selection

Bayesian methods, like the evidence and Bayes factors are nowadays frequently used to compare cosmological models (see for example [20], [21], [22], and [23]). The definition of the evidence in eq. (5)

$$p_f(\boldsymbol{d}) = \int p_f(\boldsymbol{d} \,|\, \boldsymbol{\theta}) \, p_f(\boldsymbol{\theta}) \mathrm{d}\boldsymbol{\theta}$$

tells us that $p_f(\boldsymbol{d})$ is the conditional probability of obtaining the data vector $\boldsymbol{d}$ given the model $f$. For some simple models the evidence can be calculated and a suggestive interpretation emerges [20], but in most cases the evidence of one model by itself is not very informative. Its usefulness derives from the evidence ratio used in Bayesian model selection.

If we consider another model $g$ we may compare its evidence with the evidence of the model $f$. For a consistent comparison of models within a Bayesian framework we need the joint probability $p(f \text{ and } \boldsymbol{d}) = p_f(\boldsymbol{d})\pi_f$ of model $f$ and data $\boldsymbol{d}$. Similarly for $p(g \text{ and } \boldsymbol{d}) = p_g(\boldsymbol{d})\pi_g$ of model $g$ and the same data $\boldsymbol{d}$. The $\pi_f$ and $\pi_g$ are the prior probabilities we assign to our models. Often these probabilities are chosen equal $\pi_f = \pi_g$, and the ratio of the full probabilities

$$\frac{p(f \text{ and } \boldsymbol{d})}{p(g \text{ and } \boldsymbol{d})} = \frac{p_f(\boldsymbol{d})\pi_f}{p_g(\boldsymbol{d})\pi_g} = \frac{p_f(\boldsymbol{d})}{p_g(\boldsymbol{d})} =: B_{fg} \tag{9}$$

reduces to the evidence ratio, also called Bayes factor. A $B_{fg}$ larger than unity suggests, that we should favour model[6] $f$ over model $g$.

The Bayes factor, as any result from a Bayesian analysis, explicitly depends on the prior distributions for the parameters of the models. You may have prior knowledge that allows you to specify a so called "subjective" prior [24]. Practitioners often use priors suggested by results from preceding observations or studies. Different approaches are used to motivate the so called "objective", "non-informative", or "reference" priors. Their definition can be based on the principle of insufficient reasoning, the maximum entropy principle, the invariance under transformations or scaling, or the missing information principle (see e.g. [25], [26], [27]). Kass & Wassermann [28] provide an overview and rules for selecting among these priors. For a stimulating dialogue with J.M. Bernardo on prior probabilities see [29] (don't miss the comments on this dialogue by D.R. Cox, A.P. Dawid, J.K. Ghosh and D. Lindley in the same issue). In any case, it is important to select the prior carefully, and it seems advisable to investigate the dependency of the model selection on the prior.

The calculation of the evidence (eq. (5)) can be quite challenging. Friel & Wyse [30] provide a review of different techniques. One of the first approximations for the evidence is due to Schwarz [31]. Asymptotically he arrives at the so called Bayesian Information Criterium[7] (see [33] for a detailed derivation)

$$\mathrm{BIC}(f) = -2 \sum_{i=1}^{N} \log p_f(d_i \,|\, \boldsymbol{\theta}^{\asymp}) + K \log N. \tag{10}$$

---

[6]We will comment on Jeffreys' scale in section 3; see also footnote 11.

[7]This name Bayesian *information* criterium is unfortunate, no information theory is involved here. Burnham & Anderson [32] argue that the information theoretically motivated AIC (see next section) is a Bayesian procedure with a special prior.

If we compare models, a smaller value of the BIC is better. The marginalised likelihood $p_f(d_i | \theta^{\times})$ used in eq. (10) is obtained by fixing $d_i = (x_i, y_i)$ and integrating over the remaining $\boldsymbol{d}_{[i]} = ((x_1, y_1), \ldots, (x_{i-1}, y_{i-1}), (x_{i+1}, y_{i+1}), \ldots, (x_N, y_N))^T$,

$$p_f(d_i | \boldsymbol{\theta}) = \int p_f(\boldsymbol{d} | \boldsymbol{\theta}) \, \mathrm{d}\boldsymbol{d}_{[i]}. \tag{11}$$

For a Gaussian likelihood with covariance matrix $\Sigma$ as in eq. (1), the integration can be readily performed and the marginalised likelihood is a one dimensional Gaussian with variance $\Sigma_{ii}$.

Beyond this asymptotic approach, several numerical techniques are currently used to calculate the evidence. In low dimensional parameter spaces a direct integration using standard numerical methods is sometimes possible. In cosmology a method derived from the ideas of Chib [34] has been used to estimate the evidence from a given MCMC chain [35, 36]. With nested sampling one estimates the evidence directly [37]. Several implementations of this approach are currently in use (see e.g. [38] and [39] and references therein). Kilbinger et al. [40] suggest a population Monte Carlo method to calculate the evidence. Another approach to estimate the Bayes factor is via the Savage-Dickey density ratio [41]. Comparisons of further numerical methods are discussed by [42], [43], and [30].

## 2.4 Information theoretic approach to model selection

The information theoretic approach is based on the concept of minimising the distance between the distribution of the model and the distribution of the data. We assume that some observational data $d$ is drawn at random from the true but unknown distribution with probability density $p_T(d)$. From our model $f$ and the data $\boldsymbol{d} = \{d_i\}_{i=1}^N$ we construct a predictive distribution $p_{p,f}(d)$ for a single new observation $d$. Several possibilities for such a predictive distribution exist and we will discuss the classical and the Bayesian approach. For now we assume that we know such a predictive distribution $p_{p,f}(d)$ for our model $f$ which we want to compare to the true distribution $p_T(d)$. We measure the discrepancy between the two distributions using the Kullback-Leibler (KL) divergence (see the appendix A)

$$\begin{aligned} D(p_T | p_{p,f}) &= \int p_T(d) \log \frac{p_T(d)}{p_{p,f}(d)} \, \mathrm{d}d \\ &= \int p_T(d) \log p_T(d) \, \mathrm{d}d - \int p_T(d) \log p_{p,f}(d) \, \mathrm{d}d. \end{aligned} \tag{12}$$

For model selection we rank the models $f$ and $g$ according to the value of $D(p_T | p_{p,f})$ and $D(p_T | p_{p,g})$ — the smaller the better.

### 2.4.1 Classical information theoretic approach

In the classical information theoretic approach to model selection the predictive likelihood is used as the predictive distribution. This leads to the so called Akaike Information Criterion (AIC, see [44, 45]). Applications of the AIC in cosmology are discussed in [46] and [47]. Although several definitions of a predictive likelihood exist (see [48] for a review) we follow [45] and use the marginalised likelihood eq. (11) as the predictive likelihood $p_{p,f}(d) \equiv p_f(d | \boldsymbol{\theta}^{\star})$. This is the likelihood of a new data point $d$ assuming the maximum likelihood estimate $\boldsymbol{\theta}^{\star}$ of the parameters. This is already a well defined approach if $d$ is a simple random variable. However in our regression setting we have $d = (x, y)$ and we compare the predictions of the model $f(x, \boldsymbol{\theta})$ to the observed value $y$. For each of the observed data points $d_i = (x_i, y_i)$ we know the uncertainties of the measurements entering the likelihood (compare eq. (1) and eq. (11)). But how do we calculate $p_f(d | \boldsymbol{\theta}^{\star})$ for a $d \neq d_i$? At a first glance it seems necessary

to introduce an additional model for the uncertainties. As an example we could interpolate between neighbouring values of the marginalised likelihood (eq. (11)) to determine $p_f(d \mid \boldsymbol{\theta}^\star)$. Fortunately we will see below, that this is not necessary since we evaluate $p_f(d \mid \boldsymbol{\theta})$ only at $d = d_i = (x_i, y_i)$.

Let us start with the derivation of the AIC (following loosely [49]). The first term on the second line in eq. (12) does not depend on the model $f$. The second term is the expected log likelihood for the model $f$ for all possible data

$$\eta(f) = \int \log p_{p,f}(d) p_T(d) \mathrm{d}d = \int \log p_f(d \mid \boldsymbol{\theta}^\star) \, \mathrm{d}F_T(d). \tag{13}$$

This expectation is calculated using the true cumulative distribution $F_T$. Unfortunately the true cumulative distribution $F_T$ is unknown. From the observational data $\boldsymbol{d} = (x_i, y_i)_{i=1}^N$ it is always possible to construct the empirical cumulative distribution function $F_{T,N}(d)$ as a sum of step functions (see appendix A). Then an estimate of the expected log likelihood $\eta(f)$ is given by

$$\widehat{\eta}(f) = \int \log p_f(d \mid \boldsymbol{\theta}^\star) \, \mathrm{d}F_{T,N}(d) = \frac{1}{N} \sum_{i=1}^N \log p_f(d_i \mid \boldsymbol{\theta}^\star), \tag{14}$$

where we used $\mathrm{d}F_{T,N}(d) = \frac{1}{N} \sum_{i=1}^N \delta_{d_i}(d) \mathrm{d}d$ (compare eq. (32)). Since we estimated the best parameter $\boldsymbol{\theta}^\star$ from the same dataset we use to construct the empirical distribution function $F_{T,N}(d)$, the $\widehat{\eta}(f)$ is a biased estimate of $\eta(f)$. The expected bias of $\widehat{\eta}(f)$ is

$$b(f) = \int \left( \widehat{\eta}(f) - \eta(f) \right) \mathrm{d}F_T \tag{15}$$

and the bias corrected expected log likelihood (the second term in eq. (12)) is

$$\widehat{\eta}(f) - b(f). \tag{16}$$

Clearly, this only shifts the problem from $\widehat{\eta}(f)$ to $b(f)$. Assuming that the true distribution $p_T$ is part of the family of distributions $p_f(z \mid \boldsymbol{\theta})$ and that $\boldsymbol{\theta}^\star$ is a maximum likelihood estimate, Akaike [44] shows that $b(f)$ asymptotically has the form $K/N$, with $K$ the dimension of the parameter space and $N$ the number of data points. Rescaling this approximate expression of eq. (16) by $-2N$ we arrive at the Akaike Information Criterium[8]

$$\mathrm{AIC}(f) = -2N \left( \widehat{\eta}(f) - K/N \right) = -2 \sum_{i=1}^N \log p_f(d_i \mid \boldsymbol{\theta}^\star) + 2K. \tag{17}$$

The model with the smaller value of the AIC is favoured. In Appendix B.1 we detail the bootstrap method of Konishi & Kitagawa [49] to obtain an estimate $\widetilde{b}(f)$ for the bias $b(f)$. This allows the definition of the Extended Information Criterium [49, 50]

$$\mathrm{EIC}(f) = -2N \left( \widehat{\eta}(f) - \widetilde{b}(f) \right). \tag{18}$$

The model with the smaller value of the EIC is favoured.

A comparison of eq. (17) with eq. (10) shows that the AIC and the BIC differ in how they disfavour high dimensional parameter spaces. These terms are sometimes called Ockham's razor terms. Keep in mind that the derivations of the AIC and the BIC start from different principles: the BIC starts from the evidence and the AIC from the proximity of a model to the

---

[8]We follow the convention used by H. Akaike [44].

true distribution. Several extensions and "corrections" to the AIC have been proposed (see for example [51]). A corrected AIC, better suited for smaller sample sizes, was derived by [52] (see [53] for a unifying derivation of AIC and AICc).

$$\text{AICc}(f) = -2 \sum_{i=1}^{N} \log p_f(d_i \,|\, \boldsymbol{\theta}^\star) + 2K \frac{N-K-1}{N}. \tag{19}$$

Several of the assumptions, entering the derivation of the AIC, can be relaxed and the estimates of the bias $b(f)$ can be improved (see [49] for a summary). Indeed $\boldsymbol{\theta}^\star$ need not be a maximum likelihood estimate; the asymptotic of $b(f)$ is known for Fisher consistent estimates $\boldsymbol{\theta}^\star$ and also for MAP estimates $\boldsymbol{\theta}^{\scriptstyle\times}$, as obtained from a Bayesian parameter estimation procedure. However we are only aware of the bootstrap procedure discussed by [49, 50] as a direct numerical approach to estimate $\eta(f)$ (see appendix B.1).

### 2.4.2 Bayesian information theoretic approach

In the classical approach we use the best fit marginalised likelihood $p_f(d \,|\, \boldsymbol{\theta}^\star)$ as the predictive distribution. In a Bayesian approach we use the posterior predictive distribution $p_{p,f}(d) \equiv p_{\text{ppd},f}(d)$. With the posterior distribution $p_f(\boldsymbol{\theta} \,|\, \boldsymbol{d})$ for the parameters given in eq. (4) and the marginalised likelihood $p_f(d \,|\, \boldsymbol{\theta})$ from eq. (11) we can define the posterior predictive distribution

$$p_{\text{ppd},f}(d) = \int p_f(d \,|\, \boldsymbol{\theta}) p_f(\boldsymbol{\theta} \,|\, \boldsymbol{d}) \, d\boldsymbol{\theta}. \tag{20}$$

With $p_{\text{ppd},f}(d)$ in eq. (12) we compare the posterior predictive distribution to the true distribution $p_T(\boldsymbol{d})$ using the KL-divergence:

$$D(p_T \,|\, p_{\text{ppd},f}) = \int p_T(d) \log p_T(d) \, dd - \int p_T(d) \log \left( \int p_f(d \,|\, \boldsymbol{\theta}) p_f(\boldsymbol{\theta} \,|\, \boldsymbol{d}) \, d\boldsymbol{\theta} \right) dd. \tag{21}$$

The first term does not depend on the model $f$ and the last term in eq. (21) is

$$\kappa(f) = \int \log \left( \int p_f(d \,|\, \boldsymbol{\theta}) p_f(\boldsymbol{\theta} \,|\, \boldsymbol{d}) \, d\boldsymbol{\theta} \right) dF_T(d). \tag{22}$$

We follow the strategy from Sect. 2.4.1 and insert the empirical distribution function $F_{T,N}(d)$ for $F_T(d)$ and obtain

$$\widehat{\kappa}(f) = \frac{1}{N} \sum_{i=1}^{N} \log \left( \int p_f(d_i \,|\, \boldsymbol{\theta}) p_f(\boldsymbol{\theta} \,|\, \boldsymbol{d}) \, d\boldsymbol{\theta} \right) = \frac{1}{N} \sum_{i=1}^{N} \log \left( \mathbb{E}_{\text{post}} \left[ p_f(d_i \,|\, \boldsymbol{\theta}) \right] \right), \tag{23}$$

where we expressed integral over $p_f(d_i \,|\, \boldsymbol{\theta})$ in parameter space as the expectation value $\mathbb{E}_{\text{post}}[\cdot]$ with respect to the posterior distribution $p_f(\boldsymbol{\theta} \,|\, \boldsymbol{d})$ of the parameters. We can proceed similar to the classical approach and rescale with $-2N$ to obtain the Bayesian Predictive Information Criterium[9]

$$\text{BPIC}(f) = -2N \, \widehat{\kappa}(f). \tag{24}$$

The model with the smaller value of the BPIC is favoured. As discussed in section 2.3 the value of the BPIC depends on the chosen prior. The expectation $\mathbb{E}_{\text{post}} \left[ p_f(d_i \,|\, \boldsymbol{\theta}) \right]$ can be evaluated with Markov Chain Monte Carlo (MCMC) methods. We start with one chain simulating draws from $p_f(\boldsymbol{\theta} \,|\, \boldsymbol{d})$. Only this chain is needed to calculate an estimate of $\mathbb{E}_{\text{post}} \left[ p_f(d_i \,|\, \boldsymbol{\theta}) \right]$ for each of the data points $d_i$. Contrary to section 2.4.1 we do not use a point estimate in the calculation of $\widehat{\kappa}(f)$. Therefore we think that a bias is not important in the calculation of the BPIC$(f)$. Similar biases in the closely related leave-one-out cross-validation are also considered negligible [54].

---

[9]Albeit using a different approach for the derivation, this BPIC is similar to leave-one-out cross-validation [54].

## 2.5 Other methods

A few other methods for model selection are in use. To compensate the shortcomings of ordinary $p$–values [14], posterior [55, 56] or calibrated $p$–values [57] have been suggested. Closely related to the Bayes factor is the relative belief ratio which measures the belief gained over the prior after an observation (see [58]). Seehars et al. [59] use the KL-divergence to quantify the information gained from new data sets and define the "surprise". The Deviance Information Criterium (DIC) was constructed by Spiegelhalter et al. [60] as a revised version of the AIC (see [61] and [62]). Although the DIC is popular, there is some criticism (see [63] for a summary). The derivation of the AIC [44], as sketched in section 2.4.1, assumes that the parametric models are regular[10]. For typical applications in cosmology, as given in section 3, this is the case. However models defined by multilayered neural networks are generically singular. For singular models Watanabe derived the Widely Applicable Information Criterium (WAIC, [64]). With cross-validation we split the data set. A training sample is used to determine the optimal parameters of the model and the remaining part (the validation sample) is used for estimating the discrepancy between the optimised model and the data. Then one selects the model with the smallest discrepancy (see [65] for a survey). Gelman et al. [61] compare the AIC, DIC, WAIC and cross-validation in a variety of situations. If one is interested in the estimates and the uncertainties of common parameters in nested or overlapping models, Bayesian model averaging could be a solution [66, 67]. In the introduction we mention Ockham's razor and the principle of parsimony. This can be formalised by assigning the algorithmic complexity as a unique measure to the model describing the data [68]. Typically one is not able to calculate the algorithmic complexity, but one can estimate the so called minimum description length (MDL, [69]). In its asymptotic form the MDL is similar to the AIC and BIC with yet another Ockham's razor term. The approach from complexity theory and from information theory seem to be closely related, but this is an open issue (see also [70]).

## 3 An application – the expansion history of the Universe

We illustrate these approaches to model selection by a classical example from cosmology: the accelerating expansion of the Universe as determined from redshift and luminosity measurements of supernovae [71]. The question we address is whether this data allows for a more detailed look at the expansion history of the universe and specifically if we can decide between the $\Lambda$CDM and the $w$CDM model.

A supernova type Ia (SN Ia) is a stellar explosion with a well defined luminosity [72]. In astronomy the absolute luminosity is typically specified in logarithmic units, the absolute magnitude $M_B$ in a given frequency range, here the B-band. The observed flux is measured by the apparent magnitude $m_B$ (again in logarithmic units). The distance modulus is defined as $\mu := m_B - M_B$. The redshift $z$ of the supernova or of the hosting galaxy is measured spectroscopically. In a homogeneous and isotropic universe model the distance modulus – redshift relation can be calculated. Depending on the matter content of the Universe we get

$$\mu(z, \boldsymbol{\theta}) = 5 \log_{10} d_L(z, \boldsymbol{\theta}) + 25, \tag{25}$$

with luminosity distance $d_L$ in Mpc and the redshift $z$ of the supernova. The model dependence enters through the luminosity distance $d_L(z, \boldsymbol{\theta})$ with the parameters $\boldsymbol{\theta}$. We do not pursue an exhaustive investigation of the currently fashionable cosmological models and therefore fix some of the otherwise free parameters. Consider Nicola et al. [73] and Raveri & Hu [74] for a comparison of more models using comprehensive data sets. We assume a spatially flat Universe

---

[10]A statistical parametric model is regular if its Fisher matrix is positive definite.

($\Omega_k = 0$) and choose $H_0 = 70\,\text{kms}^{-1}\text{Mpc}^{-1}$ compatible with the data from the Union 2.1 sample [75], but slightly larger than the Planck value [76]. Also if we consider only supernovae, the value of the Hubble parameter is completely degenerate with the absolute magnitude. The overall scale is given by the Hubble distance $d_H = \frac{c}{H_o} = 4.28\,\text{Gpc}$, with $c$ the speed of light. We consider two cosmological models:

1) The $\Lambda$CDM model with one parameter, the dimensionless density parameter $\Omega_m$. Since we assume a flat background we have $0 \le \Omega_m \le 1$ and $\Omega_\Lambda = 1 - \Omega_m$ for the cosmological constant term. The luminosity distance is given by (see e.g. [77])

$$d_L(z, \Omega_m) = d_H(1+z) \int_o^z \frac{\mathrm{d}z'}{\sqrt{\Omega_m(1+z')^3 + \Omega_\Lambda}}. \tag{26}$$

2) The flat $w$CDM model, with a constant equation of state $p = w\varrho$ for the dark energy component, has two free parameters $(\Omega_m, w)$. The density parameter obeys $0 \le \Omega_m \le 1$ and the cosmological term $\Omega_\Lambda = 1 - \Omega_m$ at present. The time dependence of the cosmological term is parametrised using $w$ and the luminosity distance is then

$$d_L(z, \Omega_m, w) = d_H(1+z) \int_o^z \frac{\mathrm{d}z'}{\sqrt{\Omega_m(1+z')^3 + \Omega_\Lambda(1+z')^{3(1+w)}}}. \tag{27}$$

The observational data $d_i = (z_i, \mu_i)$ are the redshift $z_i$ and the distance moduli $\mu_i$ of SN Ia. In the Union 2.1 sample we have $N = 580$ such observations from SN Ia together with an estimate $\sigma_{\mu,i}$ of the uncertainty of each distance modulus [75, 78]. Assuming a cosmological model we calculate the distance modulus $\mu_i$ given the redshift $z_i$ depending on the cosmological parameters of the model. The likelihood is the starting point for all the approaches to model selection we discussed. Similar to eq. (1) we assume a Gaussian likelihood. For the flat $\Lambda$CDM model we have

$$p_\Lambda(\boldsymbol{d}\,|\,\Omega_m) = \frac{1}{\sqrt{(2\pi)^N \prod_{i=1}^N \sigma_{\mu,i}^2}} \, \exp\left(-\frac{1}{2}\sum_{i=1}^N \frac{(\mu_i - \mu(z_i, \Omega_m))^2}{\sigma_{\mu,i}^2}\right), \tag{28}$$

with $\boldsymbol{d} = (z_i, \mu_i)_{i=1}^N$ and the distance modulus $\mu(z_i, \Omega_m)$ calculated from the model. This likelihood with a diagonal covariance matrix and model independent variances is a simplification. Our goal here is to provide an illustrative example for the different approaches to model selection. In appendix C we will discuss the statistical errors and systematic biases in more detail. For the marginalised likelihood evaluated at $d_i$ we have from eq. (11)

$$p_\Lambda(d_i\,|\,\Omega_m) = \frac{1}{\sqrt{2\pi\sigma_{\mu,i}^2}} \, \exp\left(-\frac{1}{2}\frac{(\mu_i - \mu(z_i, \Omega_m))^2}{\sigma_{\mu,i}^2}\right). \tag{29}$$

The likelihood $p_w(\boldsymbol{d}\,|\,\Omega_m, w)$ and marginalised likelihood $p_w(d_i\,|\,\Omega_m, w)$ of the $w$CDM model are defined analogously. In the Bayesian analysis we need to specify the priors. We assume a uniform distribution on $[0,1]$ for $\Omega_m$ and a uniform distribution on $[-2,0]$ for $w$. Before we turn to model selection, we estimate the parameters. For the flat $\Lambda$CDM model we obtain $\Omega_m = 0.278 \pm 0.007$. This estimate is virtually identical between the least square fit, the maximum likelihood and the MAP estimate. The error shown is the standard deviation of the posterior distribution. Similarly, we obtain $\Omega_m = 0.279 \pm 0.06$ and $w = -1.0 \pm 0.13$ in the $w$CDM model. Remember, we fixed $H_0$ and only consider spatially flat cosmological models. In figure 2 we show the data points together with the prediction of the two models, using the best fit values of the parameters respectively. In this plot the curves from the two models are lying indistinguishably on top of each other.

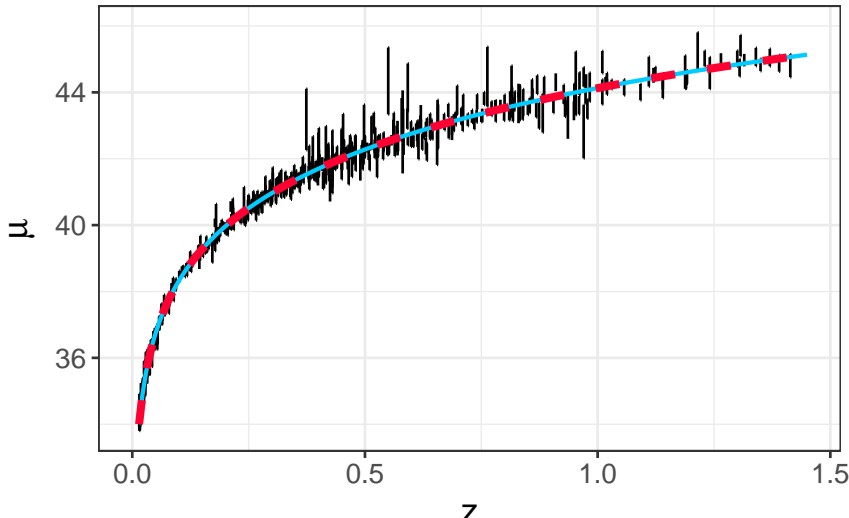

Figure 2: The redshift $z$ plotted versus distance modulus $\mu$ for the SN 1a data together with the fits for the $\Lambda$CDM model (solid blue line) and the $w$CDM model (dashed red line).

Now we apply all the methods for model selection described previously. In appendix B we give details on the numerical procedures used. First we summarise the results, later on we will put them in perspective by investigating the fluctuations of the results.

**Goodness of fit:** In the $\Lambda$CDM model we have a $\chi^2_{\text{red}}(\Lambda) = 0.971$ resulting in a $p$–value of 0.68 and in the $w$CDM model a $\chi^2_{\text{red}}(w) = 0.973$ with a $p$–value of 0.67. Neither the $\Lambda$CDM nor the $w$CDM model can be rejected.

**Likelihood ratio test:** From the maxima of the likelihoods of both models we compute the likelihood ratio $L$ and then $\lambda = -2\log L = 0.000948$ . This results in a $p$–value of $1 - G_1(\lambda) = 0.975$. Clearly we cannot reject the $\Lambda$CDM in favour of the $w$CDM model.

**Bayesian approach:** For the $\Lambda$CDM model we obtain a $\text{BIC}_\Lambda = -231.1$ and for the $w$CDM model a $\text{BIC}_w = -224.8$. Hence we should prefer the $\Lambda$CDM model. The evidence ratio is $B_{\Lambda w} = \frac{p_\Lambda(\boldsymbol{d})}{p_w(\boldsymbol{d})} = 5.45 > 1$, and again we should prefer the $\Lambda$CDM model.

**Classical information theoretic approach:** For the $\Lambda$CDM model we have an $\text{AIC}_\Lambda = -235.5$ and for the $w$CDM model an $\text{AIC}_w = -233.5$. Hence we should prefer the $\Lambda$CDM model. For the $\Lambda$CDM model we get $\text{EIC}_\Lambda = -239.3$ and for the $w$CDM model $\text{EIC}_w = -241.0$. This suggests that we should prefer the $w$CDM model.

**Bayesian information theoretic approach:** We obtain a $\text{BPIC}_\Lambda = -237.5$ for the $\Lambda$CDM model and a $\text{BPIC}_w = -237.3$ for the $w$CDM model. Therefore the $\Lambda$CDM model is preferred over the $w$CDM model.

## 3.1 Stability of the model selection

Neither using the least-square results nor with the likelihood ratio test we arrive at a definite conclusion. Both models fit the data, and we also cannot rule out $\Lambda$CDM or the $w$CDM model. In the Bayesian approach often Jeffereys' [25] scale is employed to express the numerical value of the evidence ratio $B_{\Lambda w}$ in words[11]. Hence with $B_{\Lambda w} = 5.4$ we have "substantial evidence" to support the $\Lambda$CDM over the $w$CDM. However such a "universal" scale is disputed (see e.g. [79]

---

[11]Jeffreys' scale for the evidence ratio $B$ translated to our conventions reads (see appendix B in [25]): $B < 1$: negative evidence; $1 \leq B < \sqrt{10}$: barely worth mentioning; $\sqrt{10} \leq B < 10$: substantial; $10 \leq B < 10^{3/2}$: strong; $10^{3/2} \leq B < 100$: very strong, $100 \leq B$: decisive evidence.

Table 1: A summary of the results from the Union 2.1 data set together with the dispersion estimated from the ΛCDM mock samples as described in the text.

| | results from the Union 2.1 sample | difference "$|\Lambda - w|$" | "Δ" from the mocks |
|---|---|---|---|
| $p_\Lambda$ | 0.68 | 0.01 | 0.458 [12] |
| $p_w$ | 0.67 | | |
| $p$ likelihood ratio | 0.975 | — | 0.565 [12] |
| $B_{\Lambda w}$ | 5.45 | — | 3.41 |
| $\mathrm{BIC}_\Lambda$ | -231.1 | 6.3 | 42.2 |
| $\mathrm{BIC}_w$ | -224.8 | | |
| $\mathrm{AIC}_\Lambda$ | -235.5 | 2.0 | 42.2 |
| $\mathrm{AIC}_w$ | -233.5 | | |
| $\mathrm{EIC}_\Lambda$ | -239.3 | 1.7 | 39.4 |
| $\mathrm{EIC}_w$ | -241.0 | | |
| $\mathrm{BPIC}_\Lambda$ | -237.5 | 0.2 | 42.9 |
| $\mathrm{BPIC}_w$ | -237.3 | | |

or [80]). Similarly, the mere comparison of numbers, like we did with the AIC, BIC, EIC and BPIC, is not satisfying. A scale is missing.

We do not want to propose a universal scale, which probably does not exist, but we suggest a model dependent approach to investigate the stability of our model selection. As a concrete example, consider the Bayesian information theoretic approach and the values of $\mathrm{BPIC}_\Lambda = -237.5$ and $\mathrm{BPIC}_w = -237.3$ for the ΛCDM and the $w$CDM model, respectively. To see whether this difference is important we repeatedly generate artificial data sets and calculate the $\mathrm{BPIC}_\Lambda$ for each of these data sets. This allows us to estimate the dispersion $\Delta_{\mathrm{BPIC}_\Lambda}$. Clearly the fluctuations depend on how we generate our artificial data set. We start with the Union 2.1 sample [75] and keep the redshift $z_i$ and the uncertainty $\sigma_{\mu,i}$ fixed and generate generate randomised distance moduli $\widetilde{\mu}_i$ for each of the supernovae, The $\widetilde{\mu}_i$ fluctuate around the prediction of the ΛCDM model according to

$$\widetilde{\mu}_i = 5 \log d_L(z_i, \Omega_m = 0.278) + 25 + s_i, \tag{30}$$

where $s_i$ is a random number, normally distributed with zero mean and standard deviation $\sigma_{\mu,i}$. This gives us an artificial data set $(z_i, \widetilde{\mu}_i, \sigma_{\mu,i})_{i=1}^{580}$. For one hundred of these artificial data sets we calculate the BPIC and estimate the dispersion of the BPIC using the mid-spread[13] $\Delta_{\mathrm{BPIC}_\Lambda} = 42.9$. This dispersion estimate $\Delta_{\mathrm{BPIC}_\Lambda}$ is two orders of magnitude larger than the difference between $\mathrm{BPIC}_\Lambda$ and $\mathrm{BPIC}_w$. Hence, using the BPIC we can not select one of the models. This is not a full evaluation of the fluctuations present in the model, but it helps us to assess the relevance of our results in a model dependent way (see also appendix C). The results from the Union 2.1 sample and the dispersion estimates for the $p$–values, the $B_{\Lambda w}$, BIC, AIC, EIC, BPIC are summarised in table 1. The dispersions "Δ" are always significantly larger than the observed differences between the ΛCDM and $w$CDM models. Fixing levels or using universal scales for the various criteria can hence be misleading (see also [74, 81]).

---

[12]One can show that the $p$–values obtained from these mock-samples are uniformly distributed on $[0, 1]$ and therefore the "Δ" is not really informative.

[13]The mid-spread, or inter quartile range, is defined as the difference between 75th and 25th percentiles. It is a robust estimator of dispersion. For a Gaussian distribution the mid-spread is approximately 1.34 times the standard deviation.

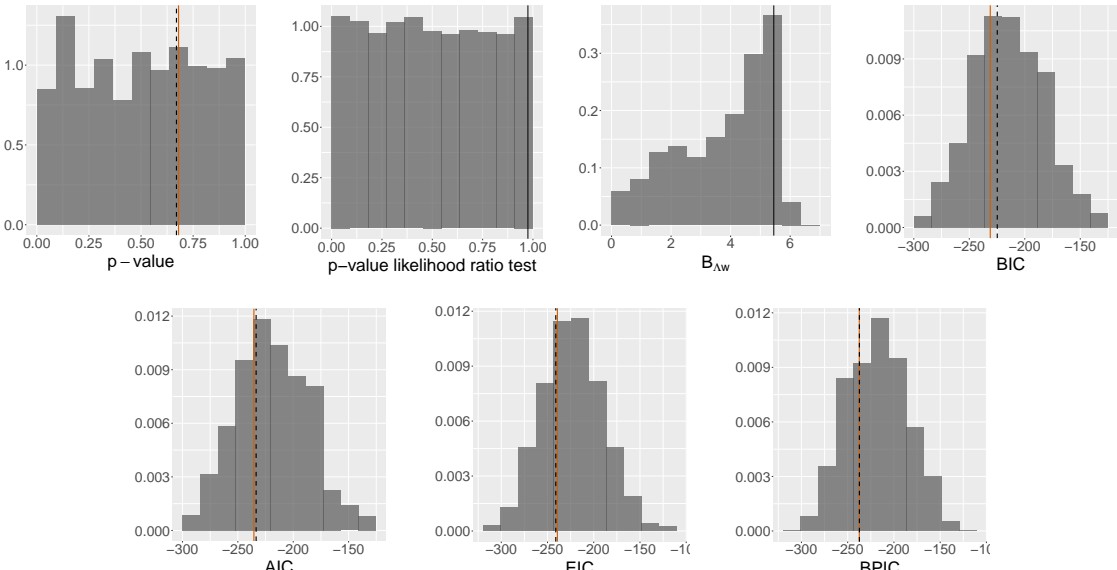

Figure 3: The normalised frequency distribution of the relevant quantities for model selection obtained from (at least) 100 ΛCDM mock samples. The vertical lines mark the values obtained from the Union 2.1 sample (compare table 1) for the ΛCDM (red) and $w$CDM model (dashed black). The $p$-value from the likelihood ratio test and the Bayes factor are already comparative quantities and only one black vertical line is shown.

We may considers the data as a realisation of a random process. Then it is quite natural to quantify the dispersions in this model dependent way. All the key figures are random variables depending on the model and the data set (considered as a random realisation). As a showcase we give the empirical distributions of all the relevant quantities for model selection within this ΛCDM mock scenario in figure 3. But one should be aware, that in classical statistics such a mock scenario is unnatural[12]. The $p$-value is considered a fixed number, only depending on the data set under investigation and the likelihood.

As mentioned in section 2.3 we study the dependence on the priors. We calculate the Bayes factor $B_{\Lambda w}$ and the BPIC for a series of priors. Still we restrict the parameters to the ranges $\Omega_m \in [0,1]$ and $w \in [-2,0]$, but in addition to the flat distribution we use Jeffreys' prior (a suitably rescaled Beta(1/2,1/2) distribution) and a series of truncated and renormalised Gaussian distributions. For the truncated Gaussian distributions we vary the width from almost flat on the intervals to strongly peaked and we use two different mean values, one is centred on the "correct" value (the MAP estimate). For all these priors the posterior distributions are very similar and the MAP estimates agree within the fluctuations. The $BPIC_{\Lambda}$ from the Bayesian information theoretic approach ranges from -237.4450 to -237.4485 for these priors. Only in the extreme cases, where we have negligible overlap between the prior and the posterior distribution, we get a value outside this range. Hence, if the prior is sufficiently broad and shows some overlap with the posterior distribution we get consistent results for the BPIC irrespective of the prior. A similar behaviour is observed for the Bayes factor $B_{\Lambda w}$.

Our conclusion comes as little surprise (see e.g. [82]): using the Union 2.1 data set we cannot decide whether the ΛCDM or the $w$CDM model should be preferred. Also keep in mind that we use a simplified ansatz for the likelihood. See appendix C for further notes.

# 4 Discussion of the methods

In physics the construction of models is guided by basic principles (conservation laws, symmetries, etc.). Adding another term, as illustrated in Fig. 1 in the introduction, is often not acceptable because one would violate these principles. For statistical applications in engineering or the social sciences this is often not a major concern. Model selection is used as a criterium to decide whether one should introduce new parameters and new dependencies. Ockham's razor suggests that one should go for the simpler model. However simplicity needs to be quantified (see Sober [83]). The dimension of the parameter space immediately comes to mind as such a measure of simplicity. But the dimension is only a rough and sometimes misleading measure of parsimony (see e.g. [11], and also compare Fig. 1). The goodness of fit, the likelihood ratio, the evidence ratio, or the KL-divergence from the information theoretic approaches are operationally well defined procedures for model selection. They allow quantitative arguments beyond mere qualitative arguments. In section 2 we describe these methods and in section 3 we apply them to a problem from cosmology. Typically one would not want to apply all of them. Neither from the mathematical definitions nor from the data analysis a clear recommendation emerges. We will now present some philosophical arguments and finally recommend the information theoretical approach. First the methods, given in Sect. 2, are briefly summarised before we critically discuss them:

- With the goodness of fit one ranks models according to their ability to fit the data points.
- With the likelihood ratio you compare the probabilities of your data given the best fitting models. Together with a predefined significance level the likelihood ratio allows you to discard a given model (your null hypothesis) in favour of the alternative model.
- In a Bayesian model comparison you use the evidence ratio to compare the joint probabilities of the models and the data. This depends on the likelihood and the prior.
- In the classical information theoretic approach you measure how good the best fitting models are at predicting new data.
- In the Bayesian information theoretic approach you measure how good the posterior predictive distributions of the models are at predicting new data.

The "goodness of fit" based on the $\chi^2_{f,\mathrm{red}}$ is sometimes used for model selection. The major shortcoming is that the $\chi^2_{f,\mathrm{red}}$ does not factor in any contributions from the false negative rate (compare appendix A). If we specify a second model and assume a Gaussian error model as well as independent sampling, the difference $\chi^2_f - \chi^2_g$ is related to the likelihood ratio as used in the likelihood ratio test.

Although the likelihood ratio test and the Bayesian model selection derive from quite different approaches towards statistical analysis, they both assume that the true model is among the considered models (see also [54]). Then you either discard the false models via tests, or you determine the most probable model. The information theoretic approach is different. There one accepts that a model is an approximation and one tries to identify the model which is closest to the true empirical distribution. This approach allows us to predict new data in the best possible way.

Similarly Wit et al. [84] discuss the following two questions (see also [54]): i) which modelling procedure will, with sufficient data, identify the true model? or ii) based on the data, which model lies closest to the true model? They conclude indecisively: asking different questions leads to different approaches for model selection. However one is able to go beyond this neutral statement. Consider the following aphorism attributed to G. Box [85]: "all models are wrong". In physics one would not use the term "wrong". Physical models have their range of applicability. We know that Newtonian gravity is failing on large scales and we assume that general relativity is failing on very small scales. However both have their range of applicability and we successfully compare their predictions with measurements and observations.

Presumably all models in physics, at least the models which may be confronted with data, are effective models (see for example the discussion of effective field theories in [86]). Hence, methods of model selection, which try to identify the true model are deceptive. We know from the outset that our models are "wrong". This is a bit nitpicking, since we know about the range of applicability of our models. Nevertheless it is advisable to respect this situation from the beginning and use the information theoretic approach. There we try to find the best approximate, not necessarily the true model. This becomes especially important in cosmology, where new observations always add to the existing data. For example new observations of galaxies are added to the already known galaxy catalogues. The Universe contains the galaxy distribution and probabilistic physical models are used to describe it (see [87]). Again, we seek the best approximating model.

Now consider another argument from the philosophy of science (see also [88]). Bayesian updating is sometimes presented as the only relevant way of plausible reasoning in science (Jaynes [89]). This would favour methods based on the Bayesian evidence and the evidence ratio for model selection. However scientists devise new models and compare them to data. Either the data supports the model or sometimes allows a rejection (falsification). This cycle has been put forward by Popper [90] and refined by Lakatosz [91]. Actually this approach seems to be too restrictive to describe the scientific growth of knowledge as outlined by Feyerabend [92] and Kuhn [93]. Laudan [94] argues that the contextual problem solving effectiveness is the key ingredient for a successful description of scientific progress (compare also with [95]). In other words, one seeks the model which offers the most effective way to describe new data. This is the idea behind the information theoretic approach.

Up to now we argued for the information theoretic approach in general but did not differentiate between the classical and the Bayesian version. As we already stated in the introduction we prefer the Bayesian approach if the prior is well specified. We do not want to repeat the arguments exchanged in the discussion of the Bayesian versus the classical approach to statistics. Perhaps the articles by Cousins [2] and Efron [96], including the comments directly following Efron's article, may serve as an introduction to this discussion. A pragmatic reconciliation is suggested by Kass [97] in his "big picture".

So far we presented methods for the comparison of models based on their ability to fit or predict observational data. There are further criteria we can and should employ to assess physical models. Independent from the observational data, new models (hopefully) make predictions and solve conceptual problems. They can be judged by their effectiveness to solve such problems [94]. This is not a quantitative endeavour, one has to present arguments for and against models, often based on the foundations of the models, or criticising the viability of the approximations used. But in the end physical models have to stand the comparison with data [98]. Then the methods of model selection we discuss come into play.

## Acknowledgements

Many thanks to all the discussants from the physical cosmology and the OPINAS seminar at LMU and the Bayes forum in Munich which helped us to improve and sharpen the argumentation. MK wishes to thank Claus Beisbart, Ulrich Schollwöck and Herbert Wagner for stimulating discussions and helpful comments.

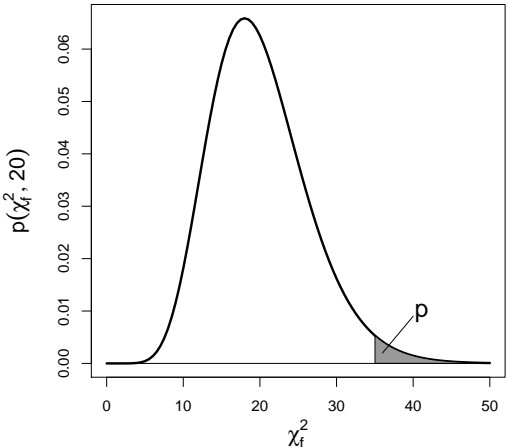
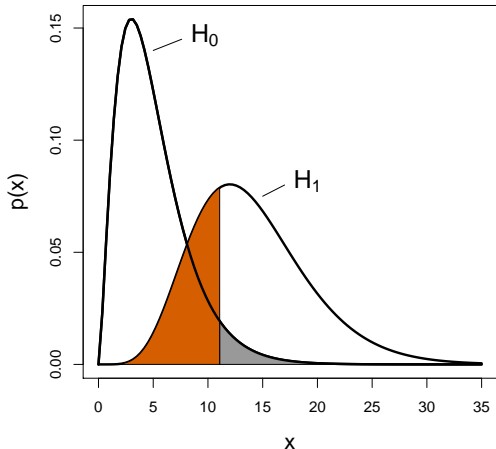

Figure 4: In the left plot the probability density $p\left(\chi_f^2, 20\right)$ of a $\chi^2$–random variable with 20 degrees of freedom is shown. The shaded area marks the probability $p = 0.02$ of obtaining a value of $\chi_f^2 > 35.0$. The right plot shows the probability densities of the null hypothesis and the alternative hypothesis. In this situation we have a false positive rate $\alpha = 0.05$ (grey) and a false negative rate $\beta = 0.32$ (red).

## A    Some results from statistics

**Statistical tests**    The theory of hypothesis tests for statistical data analysis was pioneered by K. Pearson [99]. His goal was to compare the observed frequency distribution of random events to probabilities from a model. He could show that the test statistic he developed, asymptotically follows a $\chi^2$–distribution. This approach was significantly extended by Fisher [100]. We follow the practice in physics and name the mean-square calculated in eq. (3) by $\chi_f^2$ (see e.g. [101] or [4, chap. 40]). This $\chi_f^2$ is clearly different from the test statistic used by Pearson [99]. If we make the strong assumptions that the $N$ data points used in the calculation of eq. (3) are independent and that the error is Gaussian distributed, then $\chi_f^2$ is asymptotically following a $\chi^2$–distribution with $N-1$ degrees of freedom (see also [12]).

In this situation a statistical test proceeds in the following way. Our null hypothesis is that our model with the best parameter $\boldsymbol{\theta}^*$ from the least square fit is correct. Then the $\chi_f^2$ is calculated using eq. (3). The $p$–value is given by $p = 1 - G_{N-1}\left(\chi_f^2\right)$, where $G_\nu$ is the cumulative distribution function of a $\chi^2$–distributed random variable with $\nu$ degrees of freedom. The p–value is the probability that the data may arise from the null hypothesis (see section 2.1 for more comments on the p–value). To round up the test we fix a so called significance level, typically $\alpha = 0.05$ (also called the "false positive rate" or type I error). If $p < \alpha$ we may conclude that the null hypothesis (our model) is rejected at the $\alpha = 0.05$ significance level. The left plot in figure 4 illustrates such a situation.

For the calculation of the false negative rate, specifying the null hypothesis $H_0$ alone is not sufficient [16]. We have to state an alternative model, the hypothesis $H_1$. Now assume that $H_1$ is true but $H_0$ has *not* been rejected (i.e. $H_0$ has been falsely accepted). Given $H_1$ and the false positive rate $\alpha$ we can calculate the false negative rate $\beta$ (also called the type II error) as illustrated in the right plot of Fig. 4. In the goodness of fit approach one does not specify an alternative model, the hypothesis $H_1$. Hence one is not able to quantify the false negative rate. As we see in the right plot of Fig. 4 the false negative rate $\beta$ can be quite large even for

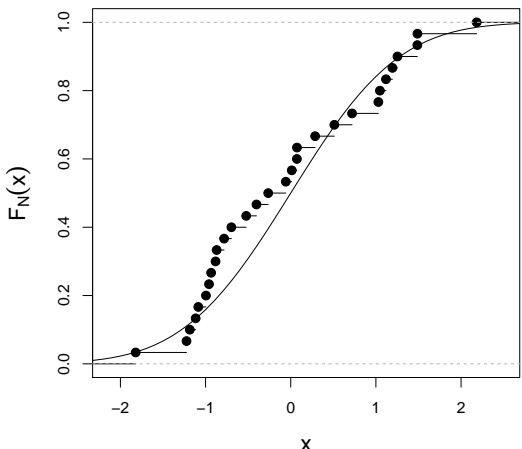

Figure 5: The cumulative distribution function of a standard Gaussian random variable together with the empirical distribution function for $N = 30$ random realisations.

small $\alpha$.

**Empirical distribution function:** For simplicity consider a real valued random variable with probability density $p(x)$ and cumulative distribution function

$$F(x) = \int_{-\infty}^{x} p(x')\mathrm{d}x'. \tag{31}$$

Consider $N$ independent random realisations $(x_1, \ldots, x_N)$ of this random variable. Then the empirical distribution function is defined as

$$F_N(x) := \frac{1}{N} \sum_{i=1}^{N} I_{[x_i, \infty)}(x). \tag{32}$$

Here $I_A(x)$ is the indicator function of the set $A$ with $I_A(x) = 1$ if $x \in A$ and zero for $x \notin A$. The theorem of Glivenko–Cantelli states that $F_N(x)$ converges for $N \to \infty$ towards $F(x)$ uniformly, this means $\|F_N - F\|_\infty \to 0$ almost surely (see [102], and compare figure 5). For example this theorem guarantees the convergence of the empirical median and quantiles. The empirical distribution function is analogously defined in higher dimensions. The half–interval is replaced by a half open rectangle stretching to infinity with the point $x_i$ marking the lower left corner.

**Kullback–Leibler divergence:** The Kullback-Leibler (KL)-divergence ( [103], also called relative entropy)

$$D(p|q) = \int p(z) \log \frac{p(z)}{q(z)} \mathrm{d}z \tag{33}$$

measures the deviation between the distribution of two random variables with probability densities $p(z)$ and $q(z)$. The KL-divergence is not symmetric in its arguments (it is not a distance). For discrete probability distributions the interpretation is straightforward. The information content in the discrete probability distribution $\boldsymbol{p} = \{p_i\}_{i=1}^{\infty}$ with $\sum_{i=1}^{\infty} p_i = 1$ is

measured by the (information) entropy [104]

$$H = -\sum_{i=1}^{\infty} p_i \log p_i, \tag{34}$$

then the KL-divergence

$$D(\boldsymbol{p}|\boldsymbol{q}) = \sum_{i=1}^{\infty} p_i(\log p_i - \log q_i) \tag{35}$$

measures the information lost, if the probability distribution $\boldsymbol{q}$ is used to approximate the true probability distribution $\boldsymbol{p}$. This characterisation carries over to the continuum. Up to a multiplicative constant the KL-divergence is a unique measure of divergence (see [105] for details).

## B Details of the implementation

We have chosen the statistical package R [106] as the basic tool for our computations[14]. The results are presented in section 3. Here you find some details about the implementation and the packages we use.

- We calculate the minimum $\boldsymbol{\theta}^*$ of the $\chi^2$ according to eq. (3) using the function `nls` from the core of R [106]. The $p$–values are calculated using the built in $\chi^2$–distribution function.
- For the maximum likelihood estimate $\boldsymbol{\theta}^\star$ we use the function `mle2` from the package `bbmle` [108]. With this $\boldsymbol{\theta}^\star$ we calculate the maximum value of the likelihood $p_f(\boldsymbol{d}|\boldsymbol{\theta}^\star)$ which we use in the computation of the likelihood ratio. And again we use the built in $\chi^2$–distribution function to calculate the $p$–value for the likelihood ratio test (see section 2.2).
- Since we only consider a one– and a two–dimensional parameter space, we are able to calculate the evidence by direct numerical integration. We use the builtin function `integrate` and an adaptive multidimensional integration routine `hcubature` from the package `cubature` [109]. The direct numerical integration gives similar results compared to the quite noisy and costly results obtained from nested sampling using the package `RNested` [110]. The BIC (eq. (10)) is calculated using a function provided in the package `bbmle` [108].
- For the classical information theoretic approach we first calculate the AIC (see eq. (17)) using functions provided in the `bbmle` package [108]. To go beyond this asymptotic result we calculate $\widehat{\eta}(f)$ according to eq. (14). With the bootstrap estimate $\widetilde{b}(f)$ of the bias $b(f)$ (see eq. (15)) we calculate the EIC($f$), see eq. (18). In section B.1 we give a detailed description of this bootstrap procedure due to Konishi and Kitagawa [49]. We use 100k bootstrap samples to estimate $\widetilde{b}(f)$.
- We prepare Markov chains with the function `metrop` from the `mcmc` package [111]. For convergence diagnostics and for tuning of the sampler parameters we employ the `coda` package [112]. Using Gelman and Rubin's convergence diagnostic [113] we see that all our chains converge after at least 1000 steps, even if we start in the extreme points of the parameter range. For the $\Lambda$CDM model we build a chain with a length of 15 Mio steps. We average the results over 15 steps and use this batched chain for the MAP estimate and to calculate the BPIC (see next point). For the $w$CDM model we build a chain with a length of 30 Mio steps and average the results over 30 steps.

---

[14]If you plan to use `python` you may consider the modules `scipy, statsmodels, arviz` and `pandas`, and see also [107] for CosmoHammer. A helpful page about python implementations for MCMC and nested sampling is maintained by Matthew Pitkin http://mattpitkin.github.io/samplers-demo/pages/samplers-samplers-everywhere.

- We calculate the BPIC($f$) as described in section 2.4.2. In the $\Lambda$CDM model we estimate for each data point $d_i = (z_i, \mu_i)$

$$\mathbb{E}_{\text{post}}\left[p_\Lambda(d_i \,|\, \Omega_m)\right] \approx \frac{1}{L}\sum_{l=1}^{L} p_\Lambda(d_i \,|\, \Omega_{m,l}), \tag{36}$$

from one Markov chain. Here $\Omega_{m,l}$ is one state from the Markov chain and $L$ is the length of the chain. Inserting this estimate of $\mathbb{E}_{\text{post}}\left[p_\Lambda(d_i \,|\, \Omega_m)\right]$ into eq. (23) we obtain $\widehat{\kappa}_\Lambda$ as an average over all the data points. Then we rescale as in eq. (24) to obtain $\text{BPIC}_\Lambda$. We proceed similarly for the $\text{BPIC}_w$.

You can download an abbreviated version of our code from https://homepages.physik.uni-muenchen.de/~Martin.Kerscher/software/modelselect/ .

## B.1   Bootstrap for $b(f)$

Before we describe the bootstrap procedure leading to the EIC [49, 50] we give a more detailed definition of the average bias. We augment the notation from Sect. 2.4.1 and express the expected log likelihood of the model $f$, the data set $\boldsymbol{d} = (x_i, y_i)_{i=1}^N$, and the cumulative distribution functions $F_T$ as

$$\eta(f; \boldsymbol{\theta}^\star(\boldsymbol{d}), F_T) = \int \log p_f(d \,|\, \boldsymbol{\theta}^\star(\boldsymbol{d}))\,\mathrm{d}F_T(d),$$

where $\boldsymbol{\theta}^\star(\boldsymbol{d})$ is the best maximum likelihood parameter obtained from the data set $\boldsymbol{d}$. The average bias from eq. (15) can be expressed as

$$b(f) = \mathbb{E}_{F_T}\left[\eta(f; \boldsymbol{\theta}^\star(\boldsymbol{d}), F_T) - \eta\left(f; \boldsymbol{\theta}^\star(\boldsymbol{d}), F_{T,N,\boldsymbol{d}}\right)\right]. \tag{37}$$

The dependence of the estimated parameters $\boldsymbol{\theta}^\star(\boldsymbol{d})$ and the empirical distribution function $F_{T,N,\boldsymbol{d}}$ on the data set $\boldsymbol{d}$ is now explicit. [49, 50] propose a bootstrap procedure to estimate $b(f)$. First generate bootstrap samples $\widetilde{\boldsymbol{d}} = (\tilde{x}_i, \tilde{y}_i)_{i=1}^N$ from the data by repeatedly drawing from $\boldsymbol{d}$ with putting back (i.e. sampling from $F_{T,N,\boldsymbol{d}}$). For each of these bootstrap samples $\widetilde{\boldsymbol{d}}$ we have the empirical distribution function $F_{T,N,\widetilde{\boldsymbol{d}}}$. The bootstrap estimate of $b(f)$, as given in eq. (37), is then[15]

$$\begin{aligned}
\widetilde{b}(f) &= \widetilde{\mathbb{E}}\left[\eta\left(f; \boldsymbol{\theta}^\star(\widetilde{\boldsymbol{d}}), F_{T,N,\boldsymbol{d}}\right) - \eta\left(f; \boldsymbol{\theta}^\star(\widetilde{\boldsymbol{d}}), F_{T,N,\widetilde{\boldsymbol{d}}}\right)\right] \\
&= \widetilde{\mathbb{E}}\left[\frac{1}{N}\sum_{i=1}^N \log p_f\left(d_i \,|\, \boldsymbol{\theta}^\star(\widetilde{\boldsymbol{d}})\right) - \frac{1}{N}\sum_{j=1}^N \log p_f\left(\widetilde{d}_j \,|\, \boldsymbol{\theta}^\star(\widetilde{\boldsymbol{d}})\right)\right].
\end{aligned} \tag{38}$$

The expectation $\widetilde{\mathbb{E}}[\cdot] \equiv \mathbb{E}_{F_{T,N,\boldsymbol{d}}}[\cdot]$ is over samples $\widetilde{\boldsymbol{d}}$ drawn from $F_{T,N,\boldsymbol{d}}$. Using $M$ such bootstrap samples $\widetilde{\boldsymbol{d}}^\alpha$, $\alpha = 1, \ldots, M$ we can estimate $\widetilde{b}(f)$ by

$$\widetilde{b}(f) \approx \frac{1}{MN}\sum_{\alpha=1}^M \sum_{i=1}^N \log\left(\frac{p_f\left(d_i \,|\, \boldsymbol{\theta}^\star(\widetilde{\boldsymbol{d}}^\alpha)\right)}{p_f\left(\widetilde{d}_i^\alpha \,|\, \boldsymbol{\theta}^\star(\widetilde{\boldsymbol{d}}^\alpha)\right)}\right). \tag{39}$$

Depending on the estimation procedure for $\boldsymbol{\theta}^\star(\widetilde{\boldsymbol{d}})$, such a bootstrap procedure can be time consuming. Konishi & Kitagawa [49] show that $\widetilde{b}(f)$ is approximating $b(f)$ for large $N$. Furthermore they propose a variance reduction scheme for this bootstrap procedure.

---

[15]Please watch where we write $\widetilde{\boldsymbol{d}}$ or $\boldsymbol{d}$.

# C  More on the data analysis

The analysis of the SN Ia data in section 3 serves as an illustrative example for the methods of model selection. To keep things simple we employ some approximations, specifically we assume a diagonal covariance matrix in the likelihood and also assume that the variances are independent from the cosmological model. Below we will try to give justice to the more complex situation.

The distance moduli $\mu_i$ of the SN Ia are calculated with a (semi) empirical relation from the observed light curve of the supernova explosion. Several parameters enter this relation (see [114] for details). In our analysis we use the $\mu_i$ provided in the Union 2.1 compilation which have been calculated with the best fit parameters. Such a uniform fitting introduces covariances between the $\mu_i$. They have been estimated and the Union 2.1 compilation comes with a non diagonal covariance matrix (see [75] and http://supernova.lbl.gov/Union/). In a full analysis we would have to include these covariances in the likelihood (compare eq. (1)). Moreover in a full Bayesian analysis we would include these fitting parameters as independent parameters and then later marginalise (compare [88]). Further error sources are photometric zero points, contamination, evolution, Malmquist bias, K-corrections, gravitational lensing, peculiar velocities, etc. (see [114]). They all contribute to the (co-)variances and have been estimated in the Union 2.1 compilation [75].

Some of these contributions to the error budget also depend on the cosmological model. For example the magnification and demagnification of high redshift supernovae by gravitational lensing depends on the structure growth, which again depends on the cosmological parameters. This lensing contribution can be estimated for some of the supernovae individually [75] but often this lensing error is estimated in a statistical sense only [115]. Then probably a self consistent treatment will be necessary if one aims for higher precision. Also anisotropies and inhomogeneities in the matter distribution influence the obeservations [116]. A careful determination of the errors will be necessary if one compares with inhomogeneous models [117–119]. Not only the distance modulus redshift relation but also the errors depend on the adopted models and have to be quantified.

Table 2: The values of the relevant quantities for model selection obtained from the Union 2.1 sample after scaling the uncertainties $\sigma_{\mu,i}$ by a factor $C$.

| C | 0.5 | 0.8 | **1.0** | 1.2 | 1.5 | 2 |
|---|---|---|---|---|---|---|
| $p_\Lambda$ | 0 | 0 | 0.684 | 1 | 1 | 1 |
| $p_w$ | 0 | 0 | 0.673 | 1 | 1 | 1 |
| $p$ l-ratio test | 0.951 | 0.969 | 0.975 | 0.980 | 0.984 | 0.988 |
| $B_{\Lambda w}$ | 23.3 | 6.82 | 5.45 | 4.54 | 3.66 | 2.41 |
| $BIC_\Lambda$ | 651.5 | -173.7 | -231.1 | -191 | -73.1 | 151.3 |
| $BIC_w$ | 657.9 | -167.4 | -224.8 | -185 | -66.8 | 157.6 |
| $AIC_\Lambda$ | 647.1 | -178.1 | -235.5 | -195.8 | -77.5 | 146.9 |
| $AIC_w$ | 649.1 | -176.1 | -233.5 | -193.8 | -75.5 | 148.9 |
| $EIC_\Lambda$ | 641.5 | -181.4 | -239.2 | -198.2 | -79.7 | 353.2 |
| $EIC_w$ | 632.1 | -185.4 | -241.0 | -200.3 | -81.2 | 352.4 |
| $BPIC_\Lambda$ | 643.9 | -180.2 | -237.5 | -197.7 | -79.3 | 145.1 |
| $BPIC_w$ | 641.9 | -180.5 | -237.3 | -197.4 | -78.9 | 145.6 |

To get a rough idea how these additional uncertainties influence our results we apply a uniform scaling factor $C$ to the $\sigma_{\mu,i}$ and then repeat the analysis from section 3. As can be seen from table 2 the values of the relevant quantities change, but compared to the dispersion estimates from the mock samples shown in table 1, the observed differences between the $\Lambda$CDM and the $w$CDM model remain small.

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
