# Peer review of "On Model Selection in Cosmology"

_SciPost Physics Lecture Notes, doi:SciPost Phys. Lect. Notes 9 (2019)_

## Round 1 · Referee Report · Mohamed Rameez (Referee 1) · 2019-2-11

Strengths

Sections 1 and 2 are clear, informative, and well written

Sections 1 and 2 serve as a good reference for statistical parameter estimation and model selection.

Many references to prior work, for the curious reader.

Weaknesses

  1. Not sufficient details about the error budget that has gone into the analyses 0f section 3.

  2. The data are being dealt with very superficially, with no discussion or acknowledgement of the sources of uncertainty, and their dependence if any on the models being compared.

  3. Section 4 makes far too many generic statements that do not necessarily follow from the discussion in Sections 1 and 2, nor apply to the analysis in section 3, with rigour.

Report

The draft is reasonably well written and clear to read. I would recommend it for publication (as a review), with the following minor concerns:

Footnote 7 on page 6: I disagree with this footnote. While the original Schwarz derivation, as well as the cited Neath and Cavanaugh derivation, indeed do not rely on information theory, it has been shown that the BIC can be derived information theoretically, by minimizing the K-L divergence (http://www.sortie-nd.org/lme/Statistical%20Papers/Burnham_and_Anderson_2004_Multimodel_Inference.pdf), just like the AIC, with the derivations only differing in the priors assigned to models with different dimensionality of parameter spaces.

The draft can perhaps be improved by providing more details about how exactly the Union 2.1 dataset and its error budget has been used in the analyses of section 3. For eg., It is clear from equations 6 and 3 that the goodness of fit test as described in the draft can be used only in the case of purely diagonal covariances. However, for supernovae, some systematics, such as dust extinction, introduce relatively large nondiagonal covariances.

In the case of the artificial datasets described in page 13 (Eq 28 and following), this is trivial, since the datasets are being generated with only diagonal covariances, but in comparing the dispersions obtained from this study with the observed difference between LCDM and wCDM, using Union 2.1, have the full covariances of the Union 2.1 catalogue been used, or are they some sort of diagonal projections?

In addition, (co)variances that have to be estimated theoretically, such as due to lensing or Malmquist bias (or peculiar velocities, which may not have been included in the Union 2.1 error budget, but has been in the JLA), are explicitly model dependent (typically estimated from LCDM predictions/simulations). If these covariances have gone into the estimators used for model selection, is a study such as this consistent and not circular? Could the fact that dispersions from the artificial datasets are 2 orders of magnitude larger than the difference between the models have something to do with this?

The draft, in section 4, proceeds to discuss aspects of philosophy of science that are far too generic/have nothing to do with the quantitative exercises carried out in the paper. For eg, Gelman and Shalizi [73], mentions that Bayesians insist on a full joint distribution of the data y and y ̅ , all possible missing uncertainties, including statistical and systematic. It’s clear that any analysis of SN1a that is Bayesian to this level of rigour will need to account for uncertainties due to the directions and redshifts of the SNe (and deviations from isotropy), especially since both are sampled sparsely, and it’s known that the local Universe has significant anisotropies at least out to z=0.067 (MNRAS, Volume 450, Issue 1, 11 June 2015).

In summary, Sections 1 and 2 serve as a good review of parameter estimation and model selection methods in statistics. Section 3 makes the jump to cosmological data analysis without sufficient detail, and section 4 makes far too many vague/generic statements that don’t seem necessarily justified based on section 3, or directly connected to sections 1 and 2.

Requested changes

  1. Add a more detailed description of the Union 2.1 error budget as included in the various estimators used on section 3, as well as the model dependence of any of those uncertainties, and its impact on model selection.

  2. Expand section 4 to tie together the various references with the content of the paper better, or cut out the vague references to various works in Bayesian inference that do not necessarily tie in with the work in the paper.

  • validity: ok
  • significance: ok
  • originality: low
  • clarity: good
  • formatting: excellent
  • grammar: excellent

Author:  Martin Kerscher  on 2019-05-16  [id 514]

(in reply to Report 1 by Mohamed Rameez on 2019-02-11)

Dear Mohammed Rameez

Thank you very much for the detailed report. We think we could incorporate your concerns (see https://arxiv.org/abs/1901.07726).

First some comments on the report

Thank you for pointing us to Burnham & Anderson. They essentially say the same as we do in our footnote: "BIC is a misnomer as it is not related to information theory" (at the beginning of their section 3). However Burnham & Anderso argue the other way round, that the AIC is a Bayesian procedure with a special prior. We mention this and give the reference.

We now mention that a marginalised likelihood can be readily calculated from a full Gaussian likelihood. But clearly this does not help with the "goodness of fit". Andrae, Schulze-Hartung, Melchior 2010 discuss in more detail the problems which are surfacing if one insists on incorporating covariances in a reduced chi^2 (we give this reference).

In the new Appendix we comment on the error budget and (co)variances entering the SN Ia sample. There we also mention the influence of anisotropies and inhomogeneous Models.

Comments on the Requested Changes

1) Our goal here is to provide an illustrative example for the different approaches to Model selection. We now explicitely mention in the application section 3 that our analysis is a simplified example with a diagonal covariance matrix and model independent variances. We added more material in a new appendix C with comments on the error budget of the supernova data. There we also comment on a full Bayesian analysis.

2) We separated the discussion of the results obtained from the supernova, from the more philosophical discussions in section 4. We added a comment in section 4 explaining why we think that these more philosophical considerations are important.

Thanks a lot Martin Kerscher Jochen Weller

---

## Round 1 · Referee Report · Paul Hunt (Referee 2) · 2019-2-25

Strengths

1) The paper is comprehensible and well-written.

2) The SN1a supernovae/dark energy models example illustrates well the use of the different model selection methods.

3) The accompanying software code promotes open science.

Weaknesses

1) No attempt is made to test or validate the methods on mock data before they are applied.

2) The presentation of the results could be better organised.

Report

This is a review of model selection methods in cosmology. A number of methods are described. To illustrate their use, two different dark energy models are confronted with SN1a supernovae observations; the authors conclude that the data cannot distinguish between the models. A particular method is recommended for philosophical reasons.

In my opinion the paper should be published. Model selection will become increasingly important in cosmology. There is a large literature on model selection, but it is scattered among many subdisciplines of science and statistics, each with their own language and notation. Therefore a review in a cosmological context will provide a useful service to the community.

The paper is well-written and the model selection methods are explained clearly, with coherent notation throughout. The choice of topics covered is reasonable. The authors are certainly well-read on the subject and give many interesting references (though perhaps some introductory textbooks on model selection could also be cited).

In addition to the usual well-known methods (likelihood ratio, Bayesian evidence, AIC, BIC etc) two more novel ones are discussed. The first is a bootstrap-based variant of AIC known as the Extended Information Criterion (EIC), although this name is not given. Its use would help to distinguish between the AIC (which is calculated using the number of parameters) and the EIC (which is evaluated using the bootstrap bias estimator). The second novel method is an information criterion scheme in which the expectation of the posterior predictive distribution is estimated using a Monte Carlo Markov chain. I suspect this method is original. The authors should either state if this is the case or provide references.

The SN1a supernovae/dark energy models example is well-chosen to reduce mathematical complexities to a minimum, as appropriate for a pedagogical guide. The uncorrelated errors mean that the marginalised likelihoods are obvious. The models only have 1 or 2 free parameters, ameliorating the computational burden which increases rapidly with the number of parameters. While the SN1a analysis could be made more elaborate (eg by including nuisance parameters for the light curve shape, colour corrections etc), this would be counterproductive - the paper is intended to teach model selection, not be the last word on SN1a supernovae or dark energy.

The authors are to be commended for releasing a software code for their work written in the statistical language R. It runs well with no bugs (although the Bayes factor computation is absent). Together with
the material in the appendixes, it means the details of the calculations are transparent and easy to replicate. Perhaps the expression
\begin{equation}
\tilde{b}\left(f\right)=\frac{1}{BN}\sum_{\alpha=1}^B \sum_{i=1}^N
\log\left\{\frac{p_f\left[d_i\,|\,\bftheta^\stari(\widetilde{\bfd}^\alpha)\right]}
{p_f\left[\widetilde{d}_i^\alpha\,|\,\bftheta^\stari(\widetilde{\bfd}^\alpha)\right]}\right\}
\end{equation}
for the bootstrap bias estimate could be included after equation (36) for clarity purposes in appendix B.1. Here the superscript $\alpha$ labels the bootstrap samples and $B$ is the number of samples.

However, I feel that the presentation of the results could be improved, and that the model selection methods should be validated using mock data (particularly the new method, which might be biased). The values of the model selection statistics and their dispersions for the SN1a supernovae/dark energy illustration are scattered throughout the text. I suggest listing them in a single table for easy reference. The dispersions are found using 100 synthetic data sets generated from a particular LCDM model. Why not also compute the mean values of the model selection statistics for these synthetic data sets? Then the quantities (actual value - mean value)/dispersion which could also be tabulated might help assess the significance of the results.

Since the model selection statistics are random variables, ideally histograms of their distributions would be plotted using synthetic data from both of the dark energy models. The actual values from the real data could be overlaid. This would help elucidate the properties of the different model selection approaches. However, I do not know if it is feasible computationally.

Given the machinery the authors have already developed, a simple performance test of the various model selection methods would be to apply them to say 1000 synthetic data sets from both dark energy models, and record the number of times each method picks the correct model.

The authors advocate the MCMC information criterion method as they favour its theoretical motivation. Since there is no consensus on model selection amongst statisticians, I take the more pragmatic view that the preferred method is the one that gives the best performance in practice. I hope that the above suggestions towards this end are helpful.

Requested changes

Major

1) Add histograms of the model selection statistics.

2) Test the performance of the model selection methods using artificial data from both dark energy models.

Minor

3) Include a table of the model selection statistics values, their dispersions and the quantities (actual value - mean value)/dispersion for the dark energy models.

4) Cite a couple of model selection textbooks.

5) Give the name EIC for the method of section 2.4.1, and address whether the MCMC information criterion method of section 2.4.2 is original.

6) Include in appendix B.1 the above formula for the bootstrap bias estimate.

Typos

p2 to name only a view -> to name only a few

p2 more parameters indeed better -> more parameters indeed better?

p3 is the "best" model -> is the "best" model?

p5 in these more general setting -> in this more general setting

p7 a single new observations -> a single new observation

p9 is not depending -> does not depend

p9 is depending on -> depends on

p13 are depending on -> depend on

p14 decide wether -> decide whether

p14 They conclude indecisive -> They conclude indecisively

p15 the best approximating -> the best approximate

p15 However scientist devise -> However scientists devise

p15 is wether this data -> is whether this data

p15 no decisive answer, too -> no decisive answer, either

p16 The left plot in figure 3 illustrate -> The left plot in figure 3 illustrates

p17 replaced by half open rectangle -> replaced by a half open rectangle

p18 the empirical distribution functions -> the empirical distribution function

p18 measures of the information lost -> measures the information lost

  • validity: good
  • significance: good
  • originality: ok
  • clarity: top
  • formatting: excellent
  • grammar: excellent

Author:  Martin Kerscher  on 2019-05-16  [id 513]

(in reply to Report 2 by Paul Hunt on 2019-02-25)

Dear Paul Hunt,

thank you very much for the detailed report. We corrected and extended the lecture note as described below (see https://arxiv.org/abs/1901.07726). Now we briefly summarize the changes we made according to your suggestions.

Major:

1) We added histograms of the relevant quantities for model selection in section 3.

2) We added more material in the application section 3 and a new appendix discussing the error budget of the supernova data.

Minor:

3) We include a table similar to the one suggested and expanded the discussion in section 3.

  1. We give references to two more textbooks in the introduction.

5) We give a reference and mention the name EIC as bootstrap version of AIC. Throughout the article we switched from tildeeta to EIC and define a BPIC instead of hatkappa. We found a variant of the leave-one-out cross validation in the review by Vehtari (2012) which is very similar to the BPIC we propose, the derivation is different.

6) We included the formula in the appendix. This makes the appendix more accessible, thank you.

  1. We corrected all the typos mentioned. Thank you very much.

Thanks a lot Martin Kerscher Jochen Weller

---

## Round 1 · Referee Report · Anonymous (Referee 3) · 2019-3-4

Strengths

  1. A good overview of many different topics
  2. Good list of references, including statistical ones and philosophy of science
  3. A nice worked out example on a case of interest to cosmologists

Weaknesses

  1. A somewhat fuzzy take-home message: at the end of the day, what should a practitioner use, and why?
  2. A lack of comparative critical analysis of the strengths and weaknesses of each method presented
  3. A lack of precision in the presentation of some quantities/concepts
  4. Some important concepts have been left out (see below for suggestions)

Report

This is an interesting paper that gives a concise and helpful first introduction to the subject. I'm not convinced however that the final recommendation is borne out by the analysis.

Requested changes

  1. p4 and throughout: The authors use "model selection" as a catch-all term. However, this is incorrect for in Frequentist terms "Hypothesis testing" has a different meaning. The authors should be more careful in distinguishing between the two, which are conceptually very different.
  2. p4: The authors should explain the conceptual difference between likelihood and posterior.
  3. p5 Below Eq. (7): "The p-value is..." this is incorrect, as Fig 3 in the manuscript shows. The p-value is the tail probability, ie, the probability of getting data more discrepant than what has been observed from the data that have been gathered.
  4. Ibid: the authors should explain that in classical hypothesis testing the goal is to rule out the null.
  5. p6: "Probably because one does not want to add a subjective touch". This is incorrect: the Bayes factor gives the update from the prior ratio to the posterior ratio for the model. One can use any model prior probability one wants.
  6. section 2.3: The authors should at least mention the Savage-Dickey density ratio as a tool.
  7. p13: "... are random variables": This is incorrect. Of course they all depend on the data realisation.
  8. The authors should discuss the maximum evidence against the simple model for nested models, as a prior-independent tool to assess the viability of additional free parameters. Ref: Sellke, Bayarri & Berger, The American Statistician, 55, 1 (2001)
  9. The conclusions of section 3 are not surprising. See e.g. https://arxiv.org/abs/astro-ph/0607496
  10. p14: "they both try to find the true model". This is incorrect. Classical hypothesis testing aims at ruling out the null hypothesis even without an explicit alternative. In Bayesian model selection, any result is provisional as it is conditional on the models being considered spanning the full range of theoretical possibilities, which is of course impossible.

  • validity: good
  • significance: ok
  • originality: ok
  • clarity: good
  • formatting: excellent
  • grammar: excellent

Author:  Martin Kerscher  on 2019-05-16  [id 512]

(in reply to Report 3 on 2019-03-04)
Category:
answer to question

Dear anonymous referee,

thank you very much for the detailed report. It helped us to make our presentation clearer. In the following we briefly summarize the changes we made according to your suggestions (see https://arxiv.org/abs/1901.07726).

1) We have clarified this at the end of subsection on "goodness of fit" and gave a reference. Probably this goes back to the debate starting in the 1930's between Fisher on one side and Neyman and Pearson on the other side about how to perform hypothesis testing. (All three of them were united against the Bayesian approach.) Neyman and Pearson advocated e.g. the likelihood ratio test which has a well defined alternative, Fisher insisted on hypothesis testing without alternative (there were also other controversial issues).

2) We added a short explanation after eq.(4).

3) We rewrote the sentence to clarify this. 3) We explain how hypotheses testing works briefly in the section "goodness of fit" and more detailed in the appendix A. See also our reply to 1) and 6).

4) We deleted this subclause.

5) We included a reference to the Savage-Dickey density ratio for estimating the Bayes factor at the end of the section on "Bayesian model selection".

6) Given the data and the likelihood, the p-value is a fixed number. This is the view from classical statistics. From the Bayesian point of view the data is a realization of a random process and the key figures are random variables. We clarified this in the section on the application and show histograms of the key figures.

7) We added references to posterior and calibrated p-values in the section on "Other methods".

8) We used the Union 2.1 as an illustrative example. We did not expect new physical insights. We stated this and included the reference. We added an appendix where we give more details on the error budget.

9) We clarified this. See also 1).

Thanks a lot Martin Kerscher Jochen Weller

---

## Round 2 · Referee Report · Paul Hunt (Referee 2) · 2019-5-27

Report

I am satisfied with the revisions to the paper and believe it to be significantly improved. I am happy to recommend it for publication.

---

## Round 2 · Referee Report · Mohamed Rameez (Referee 1) · 2019-6-3

Report

I am satisfied with the revisions introduced by the authors. I would now recommend it for publication.

---

## Round 2 · Author Response

Please see the replies to the referees.

---

## Editorial Decision

published